# Salivary Gland Adaptation to Dietary Inclusion of Hydrolysable Tannins in Boars

**DOI:** 10.3390/ani12172171

**Published:** 2022-08-24

**Authors:** Maša Mavri, Marjeta Čandek-Potokar, Gregor Fazarinc, Martin Škrlep, Catrin S. Rutland, Božidar Potočnik, Nina Batorek-Lukač, Valentina Kubale

**Affiliations:** 1Institute of Preclinical Sciences, Veterinary Faculty, University of Ljubljana, 1000 Ljubljana, Slovenia; 2Animal Production Department, Agricultural Institute of Slovenia, 1000 Ljubljana, Slovenia; 3School of Veterinary Medicine and Science, Faculty of Medicine and Health Science, University of Nottingham, Leicestershire LE12 5RD, UK; 4Institute of Computer Science, Faculty of Electrical Engineering and Computer Science, University of Maribor, 2000 Maribor, Slovenia

**Keywords:** pigs, dietary supplements, tannins, parotid gland, mandibular gland, immunohistochemistry, histology, proline-rich proteins (PRP)

## Abstract

**Simple Summary:**

Tannins have traditionally been avoided in animal nutrition due to their anti-nutritive effects. However, recent studies reported hydrolysable tannins as beneficial additives that have antimutagenic, anticarcinogenic, antidiarrheal, and antiulcerogenic effects on animals. In a study testing the inclusion of hydrolysable tannins as a potential nutritive factor to reduce boar taint in entire males, significant enlargement of the parotid glands (parotidomegaly) was observed. In this study, we aimed to determine the morphological and immunohistochemical basis for the observed parotidomegaly. We discovered that enlargement of glandular lobules and acinar area, an increased ratio between the nucleus and cytoplasm of serous cells, and increased excretion of proline-rich proteins (PRPs) were characteristic of the experimental group that received the highest dietary tannin intake. The mandibular salivary gland, on the other hand, did not show significant morphological changes among the experimental groups. This suggests increased functional activity of the parotid salivary glands as the first and most important line of defense against high dietary tannin and its potential negative effects.

**Abstract:**

The ingestion of hydrolysable tannins as a potential nutrient to reduce boar odor in entire males results in the significant enlargement of parotid glands (parotidomegaly). The objective of this study was to characterize the effects of different levels of hydrolysable tannins in the diet of fattening boars (n = 24) on salivary gland morphology and proline-rich protein (PRP) expression at the histological level. Four treatment groups of pigs (n = 6 per group) were fed either a control (T0) or experimental diet, where the T0 diet was supplemented with 1% (T1), 2% (T2), or 3% (T3) of the hydrolysable tannin-rich extract Farmatan^®^. After slaughter, the parotid and mandibular glands of the experimental pigs were harvested and dissected for staining using Goldner’s Trichrome method, and immunohistochemical studies with antibodies against PRPs. Morphometric analysis was performed on microtome sections of both salivary glands, to measure the acinar area, the lobular area, the area of the secretory ductal cells, and the sizes of glandular cells and their nuclei. Histological assessment revealed that significant parotidomegaly was only present in the T3 group, based on the presence of larger glandular lobules, acinar areas, and their higher nucleus to cytoplasm ratio. The immunohistochemical method, supported by color intensity measurements, indicated significant increases in basic PRPs (PRB2) in the T3 and acidic PRPs (PRH1/2) in the T1 groups. Tannin supplementation did not affect the histo-morphological properties of the mandibular gland. This study confirms that pigs can adapt to a tannin-rich diet by making structural changes in their parotid salivary gland, indicating its higher functional activity.

## 1. Introduction

Tannins are secondary plant metabolites found in many different plant parts. They provide an important defense mechanism [1], resulting in an astringent and bitter taste [2]. Chemically, tannins are polyphenols classified into hydrolysable (HT) and condensed tannins. HTs are esters of polyols (glucose) and various phenolic acids (gallic or hexahydroxydiphenic acid). Representatives of this group are the gallotannins, found in chestnut wood, which, together with oak and eucalyptus, are the most commonly used plant for commercial tannin extracts [3].

The biological effects of tannins vary from beneficial (antioxidant [4,5,6], antimutagenic [7,8,9,10], anticarcinogenic [11,12], antidiarrheal [13,14], and antiulcerogenic [15,16,17]) to harmful (hepatotoxic [18], nephrotoxic, antinutritive [19], and carcinogenic [20]) [1,21]. The effect depends on their chemical structure, the concentration of tannins indigested, and the animal species ingesting them.

Tannins are generally considered antinutritional factors that reduce palatability, digestibility, and protein utilization of feeds, negatively affecting growth performance [3]. As thoroughly reviewed by Caprarulo et al. (2021), the beneficial effects of tannin supplementation in pig farming are related to their antimicrobial, antioxidant, and radical scavenging, anti-inflammatory activities, and immune status. However, the underlying mechanism is not fully understood [22]. Interestingly, in Iberian pigs, known for their high resistance to tannin-rich diets, even high amounts of tannins allow for fattening of the boars [23]. Positive effects of polyphenols on reproductive performance, antioxidants, and overall health status were reported in sows during early gestation [6]. Protein digestibility is further affected by tannins and was initially associated with the binding of tannins to digestive enzymes and ingested proteins [24]. However, later studies suggested that the binding of tannins to proteins within the saliva and intestinal mucous mainly causes this anti-nutritive effect [2]. Due to tannin hydrolysation in the digestive tract, phenolic acid is secreted and absorbed into the bloodstream through the intestinal mucosa, potentially leading to acute hepatic necrosis and renal failure [25,26]. However, animals have developed specific defense mechanisms against tannins, such as selective feeding [27]. In addition, animals have acquired the ability to secrete high amounts of proline-rich proteins (PRPs) in saliva. These bind to tannins, preventing their interaction with other proteins [28]. According to their role, PRPs are characterized as basic PRPs (BPRP) and acidic PRPs (APRP). Based on their high affinity for tannins, the BPRPs represent the major defense mechanism against tannins in animals and humans [29,30]. By forming tannin-protein complexes, they protect the gastrointestinal (GI) mucosa because the affinity of PRPs for tannins is high. This allows PRPs to serve as the first line of defense against dietary tannins. Furthermore, PRPs prevent tannins from binding to other proteins such as digestive enzymes or proteins in the GI mucosa.

The salivary glands, particularly the parotid gland and, to a lesser extent, the mandibular gland, have evolved as an adaptive mechanism against high amounts of tannins in the diet. They produce APRPs (mandibular gland) or both APRPs and BPRPs (parotid gland) and respond to high tannin intake with hypertrophy and increased PRP secretion [31].

Pigs can neutralize high amounts of tannins in their diet and adapt relatively quickly to a tannin-rich diet, allowing them to consume tannin-rich diets without negative effects compared to other mammalian species [1,32,33]. Tannins in small amounts are now commonly used in pig diets for their antiparasitic, antimicrobial, and antiviral effects and as a supportive treatment for diarrheal diseases [3,34,35]. Since HT can affect microbial activity in the hindgut, tannin wood extract supplementation has recently been tested as a potential feed additive to control boar taint [36,37,38] when fattening uncastrated male pigs. Differences in parotid [36] and mandibular [38] gland size were observed in pigs fed with high tannin supplementation. This study aimed to evaluate the dose-dependent effect of HT inclusion in the diet on the morphological structure of the salivary glands (parotid and mandibular gland) and the expression of PRPs (APRPs and BPRPs) at the histological level.

## 2. Materials and Methods

### 2.1. Study Design and Sampling Procedure

The samples used in the present research originate from a study previously described by Čandek-Potokar et al., evaluating the effect of tannins on growth performance and skatole production [36]. The study was not subject to ethical protocols according to Directive 2010/63/EU (2010) 119. Approved food additives were used (European Union Register of Feed Additives, 2013). Briefly, 24 crossbred pigs were assigned to four treatment groups within a litter and housed individually in 1.3 × 2.1 m pens with slatted walls. Initially receiving the same treatment, from 60 kg live weight onwards, the control group (T0) was fed a commercial feed mixture (13.2 MJ/kg, 15.6% crude protein). The experimental groups T1, T2, and T3 were offered the same feed mixture supplemented with 1%, 2%, and 3% chestnut wood extract Farmatan^®^ (Tanin Sevnica kemična industrija d.d., Sevnica, Slovenia), respectively. The wood extract is rich in HT, mainly gallotannins [35,38]. The concentration of total phenols in the tannin extract was determined using the Folin-Ciocalteau calorimetric method and showed a content of 43.6%, expressed as gallic acid equivalents [39]. The ingredients and chemical composition of the experimental feed mixture had been published previously [36]. All animals were housed individually with *ad libitum* access to feed and water. At the average weight of 122 ± 10 kg and at 193 days old, the boars were slaughtered in one slaughter batch at a commercial abattoir using the standard procedure (CO_2_ stunning and vertical bleeding). The day after slaughter, the salivary glands (parotid and mandibular) were dissected from the cooled carcasses and preserved for histological analyses.

### 2.2. Histological Preparation and Measurements of the Specimens

The collected parotid and mandibular gland specimens were fixed in 10% buffered formalin and embedded in paraffin using a Leica TP1020 automatic tissue processor (Leica Biosystems, Deer Park, IL, USA). These were then cut into 5 µm thick tissue sections using a manual microtome (Leica Biosystems, Deer Park, IL, USA) and transferred onto the smooth surface of a warm water bath (40 °C) and from there onto glass microscopic slides. The samples were stained with trichrome Goldner stain (Masson-Goldner staining kit, Merck, Darmstadt, Germany) according to standard procedures. Briefly, after initial deparaffinization and rehydration, nuclei were stained with Fe hematoxylin for 2 min. Samples were washed under tap water for 10 min then rinsed in 1% acetic acid for 30 s, stained in azophloxin solution for 10 min, and subsequently rinsed in 1% acetic acid for 30 s. Samples were then incubated in tungstophosphoric acid orange G solution for 1 min, rinsed with 1% acetic acid for 30 s, incubated in light green SF solution for 2 min, and finally rinsed in 1% acetic acid for 30 s. Samples were dehydrated in increasing concentrations of alcohol and cleaned with xylene, and mounted with Neomount (Merck Milipore, Darmstadt, Germany). They were then examined microscopically using a Nikon FXA microscope (Nikon instruments Europe B.V., Badhoevedorp, TheNetherlands). In addition, the parotid gland samples were stained with immunohistochemical methods, using a rabbit anti-PRB2 antibody against basic PRPs (Abgent, San Diego, CA, USA) at a ratio of 1:100 in phosphate buffered saline (PBS) or a goat anti-PRH1/2 antibody against acid PRPs (Santa Cruz biotechnology, Dallas, TX, USA) at a ratio of 1:25 in PBS.

We measured the average area of 30 serous acini per sample (at 40× magnification), the average area of 20 ducts per sample (at 20× magnification), the average area of all of the lobules within the sample (at 2× magnification), the average area of 20 serous cell nuclei per sample (at 40× magnification), and the average area of 20 serous cells per sample (at 40× magnification) of the histological samples of the parotid and mandibular glands with trichrome Goldner staining. Additionally, the number of serous acini was counted on eight microscopic fields (at 20× magnification), as was the number of ducts per lobule in eight different lobules. In the mandibular glands, additional measurements of the average area of 30 mixed acini (at 40× magnification), the average area of 20 mucous cells per gland (at 40× magnification), and the average area of 20 mucous cell nuclei per gland (at 40× magnification) were conducted. Additionally, the mucous and mixed acini were counted on eight microscopic fields (at 20× magnification).

### 2.3. Image Analysis of Immunohistochemical Preparations

The samples immunostained with antibodies against basic PRB2 or acidic PRH1/2 were further analyzed using RGB (red, green, blue) color space analysis. R, G, B, channel properties, and distance to black were evaluated for each pixel within the images from five different microscopic fields per sample at 10× magnification. The mean and standard deviation were calculated for each of the properties.

### 2.4. Statistics

The results from the stereological measurements were statistically analyzed using Graph Pad Prism 8 (Graph Pad Prism Software Inc., San Diego, CA, USA). For statistical analysis, data distribution was first confirmed using the Shapiro–Wilk test. The effects for each treatment group were analyzed using one-way ANOVA, and mean values were compared using Tukey’s multiple comparison test. A level of *p* < 0.05 was used for statistical significance.

## 3. Results

### 3.1. Histomorphology and Histometry of Parotid Salivary Glands

The areas of the lobules and acini could be examined easily in the samples stained with trichromatic Goldner’s stain (Figure 1) due to the green staining of the connective tissue. A significant effect related to tannin supplementation was observed for most of the measured histological structures within the parotid gland (Table 1). The areas of the lobules, serous acini, and serous cells were significantly increased in the T3 compared to the control (T0) and other experimental groups, whereas the number of ducts decreased overall. Moreover, the linear trend in the number of ducts was significantly larger in lower tannin diet groups but increased as tannin extract amounts increased. The increase was most pronounced in the lobular area (1.6-fold increase in T3 compared to T0) and area of serous cells (1.2-fold increase in T3 compared to T0). In contrast, the number of ducts per lobule showed an overall decrease in groups with higher tannin supplementation, similar to the calculated ratio between the area of serous cell nuclei and cytoplasm, exhibiting a statistically significant lower ratio in the T3 group (*p* = 0.0171; linear trend = 0.0047; Figure 2).

### 3.2. Immunohistochemical Assessment of the Parotid Salivary Gland

The parotid gland samples were examined immunohistologically to determine the potential increase in PRP content. Visual assessment of the staining with antibodies against basic PRPs (anti-PRB2 antibody, Figure 3) showed different staining intensities in the glandular acini where PRB2 is normally present. The control group showed a lower staining intensity, whereas the T3 group showed the highest intensity. Measurements of staining intensity were performed in RGB color space and the distance to black to quantify the differences more precisely. The distance to black, R, G, and B parameters had 1.2-, 1.0-, 1.2-, and 1.4-fold decreases, respectively (*p* < 0.001) in the T3 group compared to the control (T0). The T3 group represented the highest intensity of staining with PRB2 and therefore contained the highest protein content in the glandular acini (Figure 4).

The increasing effect of tannin supplementation was observed by a linear trend, showing an inverse correlation with all experimental parameters (Figure 4).

Figure 5 shows representative photomicrographs of immunohistochemical staining with antibodies against acidic PRPs (anti-PRH1/2 antibodies). Visual assessment of staining using these antibodies showed uniform staining across the acini areas where PRH1/2 was localized.

On the images stained with PRH1/2, a 1.1-fold decrease was observed in all of the measured parameters (distance to black, R, G, and B) for the T1 (*p* < 0.001) group, whereas the other groups showed no differences compared to the control (T0) group (Figure 6).

### 3.3. Histomorphology and Histometry of Mandibular Salivary Glands

The histological structure of the mandibular salivary glands from all groups was consistent with the normal structure in pigs [40], as shown in Figure 7.

Compared to the parotid gland, the effect of tannin supplementation on the measurements of histological structures in the mandibular gland was less pronounced (Table 2). Nevertheless, the measurements of the lobule areas showed a 1.6-fold increase in T2 (*p* > 0.0001) compared to T0. Surprisingly, it was only 1.2-fold higher in the group with the highest tannin supplementation (T3; no statistically significant difference, *p* > 0.1). A tendency toward a decreased area of mucous acini (*p* = 0.0794) and excretory ducts (*p* = 0.1154) was observed with increased tannin supplementation (shown by a significant linear trend, *p* < 0.04).

No differences were observed for the nucleus to cytoplasm ratios of serous cells (linear trend = 0.3506, nonlinear trend = 0.6919) and mucous cells (linear trend = 0.1607, nonlinear trend = 0.7783) between the experimental groups (Figure 8).

## 4. Discussion

The salivary glands examined here originated from a previously published study [36] in which significant enlargement of the parotid glands was observed. Here, we further assessed morphological characteristics of salivary glands and showed that the observed parotidomegaly, especially in the T3 group (highest tannin intake), occurred as a response to cellular hypertrophy.

Previous studies have reported parotidomegaly in different animal species. Cappai et al. reported parotidomegaly in several studies on pigs receiving 70% [33], 70% or 50% [1] and 50% [41] acorn in their diet. Morphometrically, parotidomegaly was determined by measuring the salivary gland’s mass and size (height and width) and was compared among different groups by calculating the enlargement factor (EF). EF was shown to be significantly increased in the group of pigs fed with a high acorn diet. Furthermore, they showed a dose-response of parotidomegaly (presented as EF) to daily tannin intake, showing a progressive increase of parotid mass correlated with higher tannin intake [1]. Parotidomegaly in response to a tannin-rich diet, has also been described in mice, showing an increase in acinar area and the perimeter of the parotid salivary gland [42].

Histological evaluation of tannin-induced changes in the parotid salivary gland has been addressed previously. However, we measured several different structures in both parotid and mandibular salivary glands and performed a statistical analysis comparing the tannin effects on the structure, which have not been published before. Morphometric evaluation of the parotid gland showed a statistically significant increases in acinar and lobular areas in the group receiving 3% tannin supplementation (T3) compared to the control group (T0). In addition, comparing all experimental groups, we observed a significant increase in the linear trend for these parameters and the number of excretory ducts per lobule. Importantly, the increases in the parameters mentioned above probably occurred due to the growing cytoplasmic area of serous cells (observed as a lower nucleus to cytoplasm ratio) in experimental groups, especially in the T3 group. Similar changes were reported by Cappai et al., who observed higher secretory activity in acinar cells of the parotid gland from pigs receiving a high tannin diet. They presented a higher activity level by the presence of nuclei, euchromatin, and vacuolized cytoplasm in acinar cells [33,41]. Moreover, they confirmed this by calculating the cytoplasm/nucleus ratio, which was significantly higher in the acorn-fed group. As previously discussed, an increase in acinar area was also confirmed in mice, showing the comparable effect of tannins on parotid gland histology in different species. Importantly, the morphometric characteristics described above classify the observed parotidomegaly as “hypertrophia” rather than “hyperplasia,” which is defined as an increase in cell number, which was not observed in our study nor previous studies [41]. A similar reaction within acini was also observed in other studies on pigs [42] and rats [43]. Comparable to our results, these studies reported the enlargement of serous acini after feeding tannins to the animals and linked this enlargement to the higher secretory activity of acinar cells and the production of salivary proteins [42,43]. Enlargement of salivary gland acini seems to be a common response to external stimuli promoting higher production of saliva or salivary proteins, as shown in the case of *Leishmania infantum* infection in dogs [44]. Furthermore, we did not observe significant changes in the duct cell area, which was in line with previous studies, confirming the importance of acinar cells in response to a tannin-rich diet [41,45].

We further investigated the link between the observed increase in secretory activity in serous cells and the amount of basic or acidic PRPs in the parotid gland. PRP proteins act as buffers against the negative effects of tannins. They are thought to have three functions: (i) they are important in maintaining oral health and protecting the GI tract mucosa by binding tannins and producing tannin-protein precipitates; (ii) PRPs, especially basic PRPs, are composed of a high amount of the non-essential amino acid proline that can selectively bind tannins due to its open structure and physiochemical properties, preventing precipitation of essential amino acids by tannins; and (iii) complexes of tannin-PRP proteins reduce the activity of digestive enzymes and thus the utilization of nutrients [46,47]. Using immunohistochemistry, we showed significant changes in staining intensity with PRB2 antibodies against basic PRPs for the T3 group and a significant increase in the linear trend for all parameters. This corresponds with the observed high secretory activity of acinar cells in groups receiving high tannin supplementation. However, staining with anti-PRH1/2 antibodies targeted against acidic PRPs showed non-significant color intensity, possibly due to species reactivity. Other studies have described the association between increased PRP secretion from the parotid gland and parotidomegaly when animals are fed tannins [1,33,42,48]. In pigs, PRPs were found predominantly in the parotid gland, confirming the results of our study in which we observed higher levels of basic PRPs in the parotid gland. Increased synthesis of PRP proteins or proline by the adenomeric cells in the parotid gland of pigs fed a high acorn diet resulted in increased PRP secretion in saliva [1]. Higher excretion of PRPs has also been described in laboratory animals (mice) and other mammalian species [2,49] and is associated with the expression of PRP genes [50,51].

In addition to the measurements performed on parotid salivary glands, histological evaluation of the mandibular gland was also performed. The mandibular glands showed no significant changes in the mass or parameters measured between the experimental groups. A significant change was only observed in the lobular area, where the T2 group showed an increase in size. However, nonlinear or linear trends toward the decrease in areas of lobuli, mucous acini, and ducts were observed in the mandibular glands. Our results are consistent with the previously described results by Bee et al., who also observed decreased mandibular gland mass when feeding pigs with 15- or 30 g/kg of HT daily. This decrease showed the indirect influence of the mandibular gland on pheromone-related androstenone production, where high tannin content in the feed significantly reduced the mandibular gland mass and androstenone concentration in fatty tissue [38]. Tissue samples of examined salivary glands showed no pathological changes, which suggests that the salivary gland adaptation to higher concentrations of tannins in the diet is a physiological response. Together with the previously published effect of HT on growth performance and skatole concentration by Čandek et al., these results indicate the positive effects of tannins on boars [36]. Overall, the addition of the tannin concentrations used appears beneficial during boar production, as previously indicated by Caprarulo et al. 2021 [22].

From the data observed in our study, we conclude that adding tannin extract to the diet affects the activity of the salivary glands, especially the parotid gland. The tendency for higher PRP excretion suggests an important role of saliva as a first-line defense mechanism to protect the mucosa of the GI tract.

## 5. Conclusions

In summary, we investigated the effect of a tannin-rich diet on the morphological changes in salivary glands in pigs based on the distinct parotidomegaly observed. We report significant influences on the morphometric characteristics in the group of pigs fed with the highest tannin supplementation (T3 group) and linear trends towards increases in lobular and acinar area and the number of ducts per lobule. Moreover, using an immunohistochemical approach, we showed a significant increase in the level of basic PRPs in the T3 group and acidic PRPs in the T1 group in the parotid salivary gland. On the other hand, mandibular salivary glands showed decreases in measured parameters, suggesting the parotid salivary gland is the main defense mechanism against high tannin concentrations in feed.

## Figures and Tables

**Figure 1 animals-12-02171-f001:**
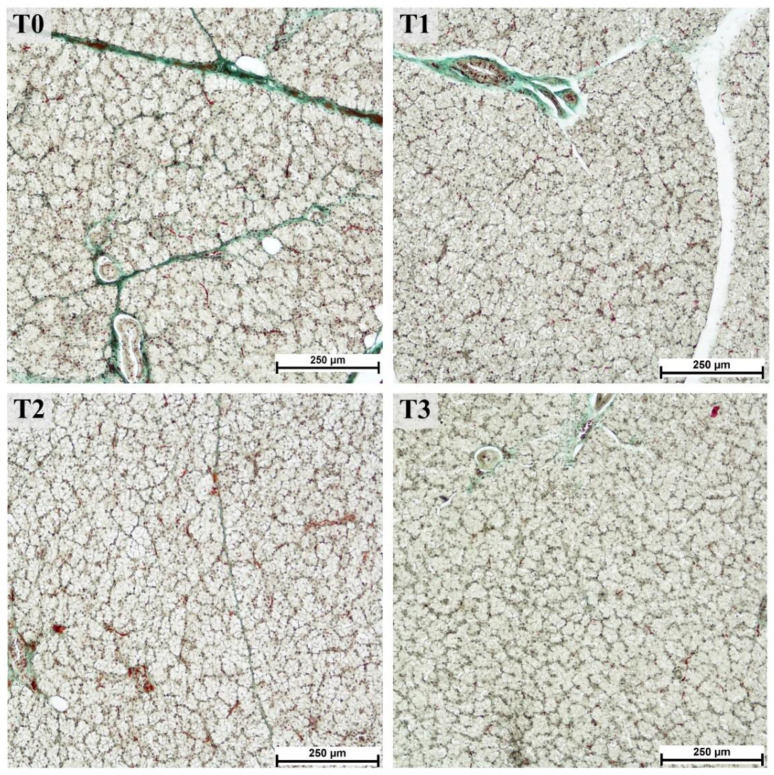
Serous acini in parotid glands from fattening boars fed diets supplemented with different tannin levels. Group fed with the standard mixture (T0; control group), and the standard mixture supplemented with 1% (T1), 2% (T2), or 3% (T3) tannin-rich extract Farmatan^®^. Goldner’s Trichrome method, 10× magnification.

**Figure 2 animals-12-02171-f002:**
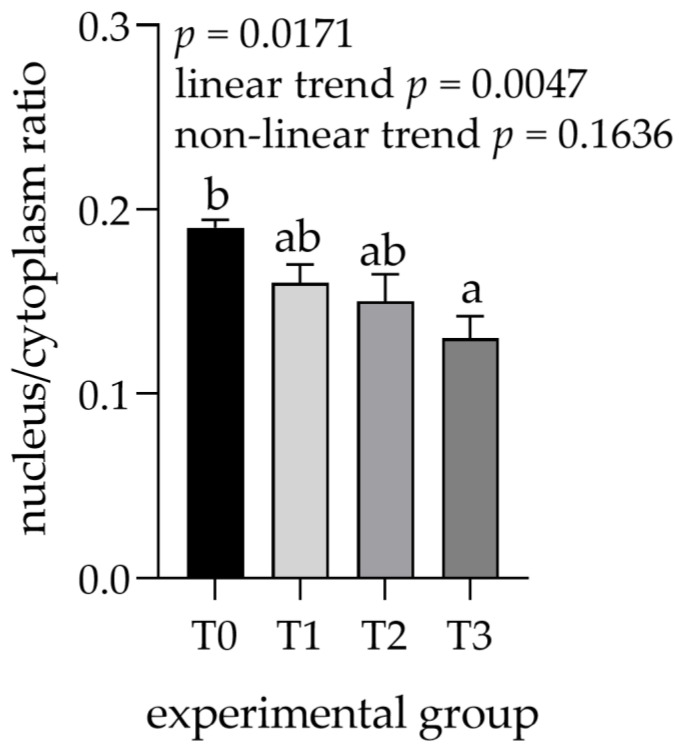
Nucleus to cytoplasm ratio measured in parotid glands. Groups were fed with a standard mixture (T0; control group), or the standard mixture supplemented with 1% (T1), 2% (T2), or 3% (T3) tannin-rich extract Farmatan^®^. ^a,b^ Different superscript letters indicate significant differences (*p* < 0.05).

**Figure 3 animals-12-02171-f003:**
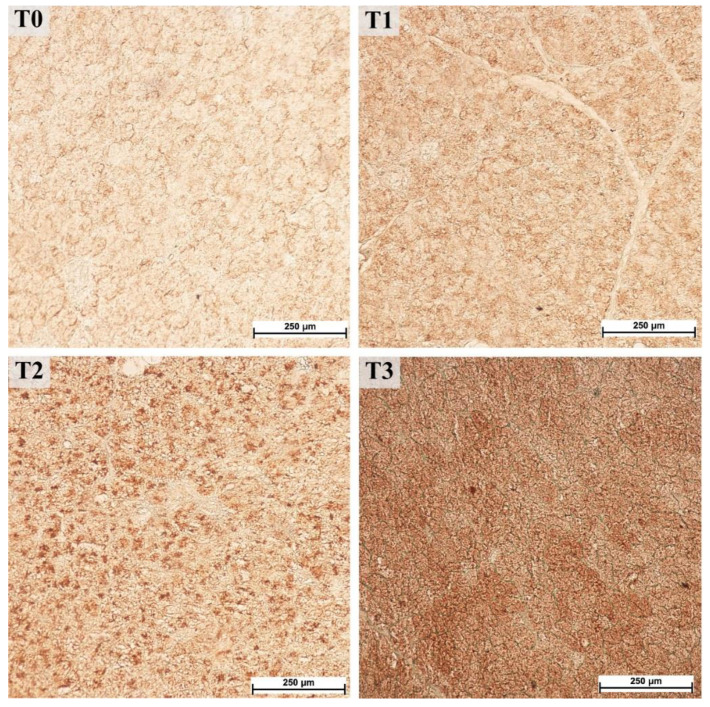
Immunohistochemical staining with antibodies against basic proline-rich proteins (PRB2) in the parotid gland, 40× magnification. Groups were fed with the standard mixture (T0; control group), or the standard mixture supplemented with 1% (T1), 2% (T2), or 3% (T3) tannin-rich extract Farmatan^®^.

**Figure 4 animals-12-02171-f004:**
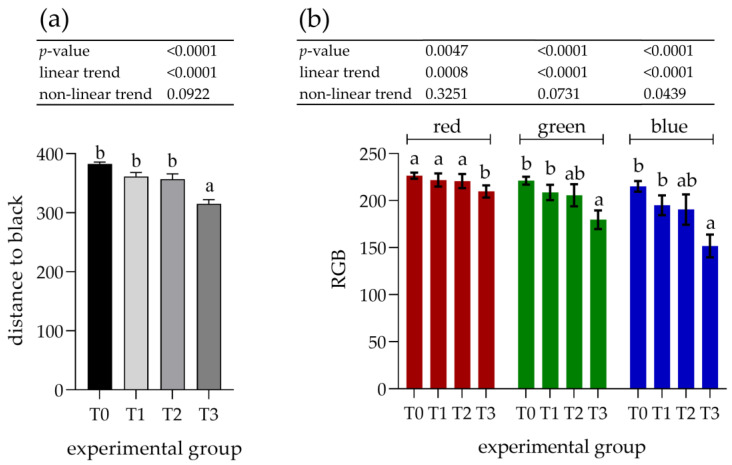
Assessment of immunohistochemical staining with antibodies against basic proline-rich proteins (PRB2) in the parotid salivary gland. (**a**) Distance to black; (**b**) RGB channel (red, green, blue). Groups were fed with the standard mixture (T0; control group) or standard mixture supplemented with 1% (T1), 2% (T2), or 3% (T3) tannin-rich extract Farmatan^®^. Values represent mean ± SEM for each channel. ^a,b^ Different superscript letters indicate significant differences (*p* < 0.05).

**Figure 5 animals-12-02171-f005:**
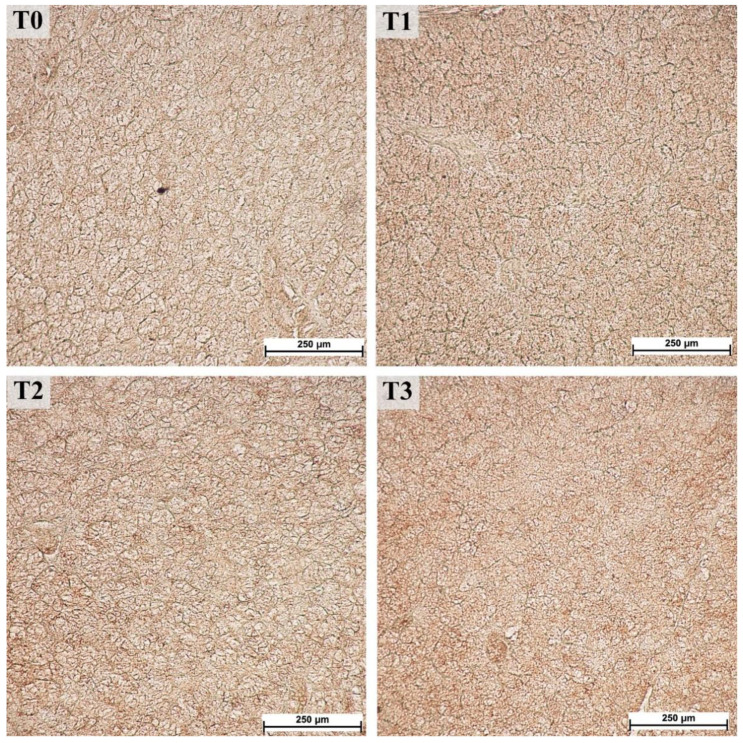
Immunohistochemical staining with antibodies against acid proline-rich proteins (PRH1/2) in the parotid gland, 40× magnification. Groups were fed with the standard mixture (T0; control group) or standard mixture supplemented with 1% (T1), 2% (T2), or 3% (T3) tannin-rich extract Farmatan^®^.

**Figure 6 animals-12-02171-f006:**
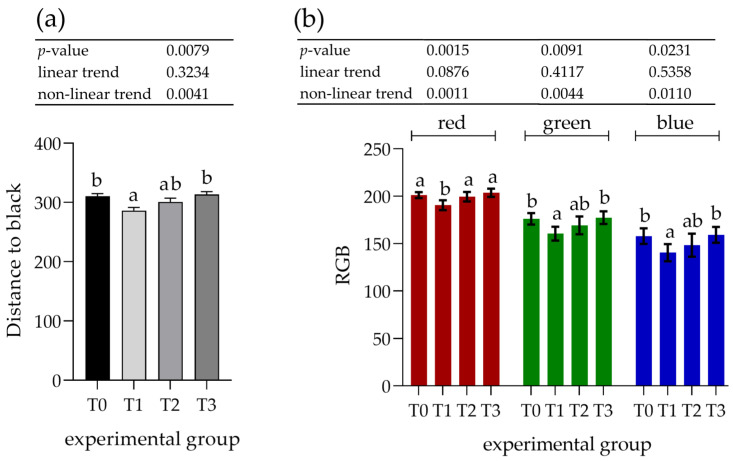
Assessment of immunohistochemical staining with antibodies against acidic proline-rich proteins (PRH1/2) in the parotid salivary gland. (**a**) Distance to black; (**b**) RGB channel (red, green, and blue). Groups were fed the standard mixture (T0; control group) or standard mixture supplemented with 1% (T1), 2% (T2), or 3% (T3) tannin-rich extract Farmatan^®^. Values represent mean ± SEM for each channel. ^a,b^ Different superscript letters within a row indicate significant differences (*p* < 0.05).

**Figure 7 animals-12-02171-f007:**
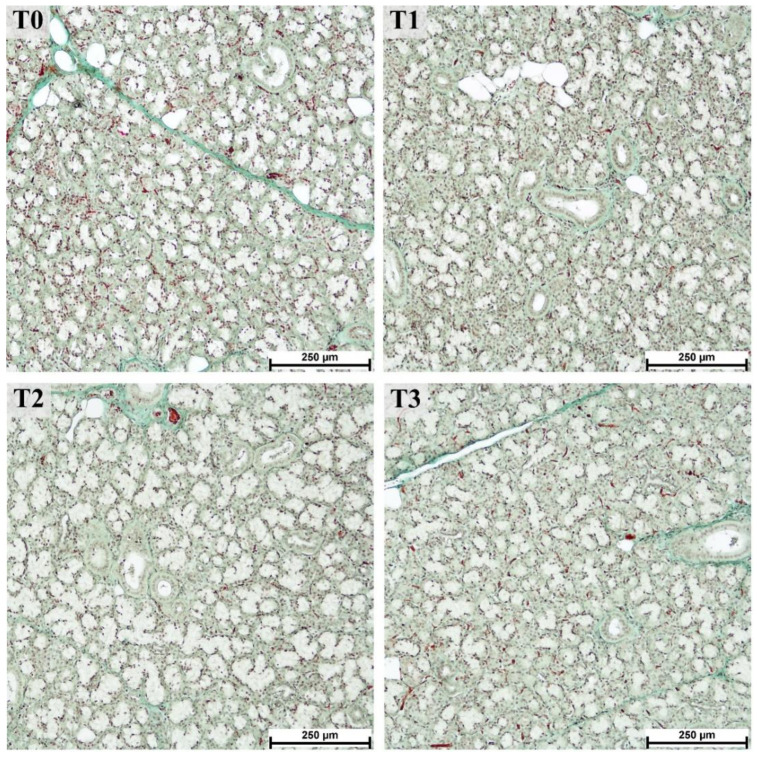
Mucous and mixed acini of the mandibular gland from fattening boars fed diets supplemented with different tannin levels. Groups were fed with the standard mixture (T0; control group) or standard mixture supplemented with 1% (T1), 2% (T2), or 3% (T3) tannin-rich extract Farmatan^®^. Goldner’s Trichrome method, 10× magnification.

**Figure 8 animals-12-02171-f008:**
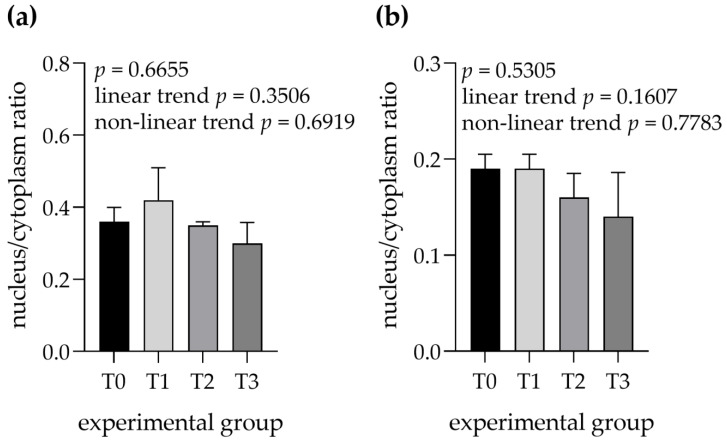
Nucleus to cytoplasm ratios were measured in (**a**) serous cells and (**b**) mixed cells within mandibular glands. Groups were fed with the standard mixture (T0; control group) or standard mixture supplemented with 1% (T1), 2% (T2), or 3% (T3) tannin-rich extract Farmatan^®^. Values represent mean ± SEM.

**Table 1 animals-12-02171-t001:** Morphological traits of parotid glands from fattening boars fed diets supplemented with different tannin levels.

Experimental Group	T0	T1	T2	T3	*p*-Value	Linear Trend	Nonlinear Trend
Area of lobules [mm^2^]	1.8 ^a^ ± 0.2	1.7 ^a^ ± 0.1	2.6 ^ab^ ± 0.4	2.9 ^b^ ± 0.9	0.0015	0.0003	0.2385
Area of serous acini [µm^2^]	1084 ^a^± 31	1038 ^a^± 20	1026 ^a^ ± 151	1342 ^b^ ± 13	0.0019	0.0160	0.0035
Number of serous acini *	55.1 ± 0.9	51.8 ± 1.6	54.5 ± 6.7	46.9 ± 2.0	0.1864	0.1132	0.1169
Area of ducts [µm^2^]	2865 ± 193	2822 ± 174	2882 ± 231	2855 ± 247	0.9469	0.8442	0.8535
Number of ducts per lobule	12.4 ± 1.5	12.9 ± 1.1	12.5 ± 1.5	8.2 ± 0.8	0.0876	0.0155	0.4923
Area of serous cells [µm^2^]	155 ^a^ ± 5.3	166 ^a^ ± 6.2	165 ^ab^ ± 10.4	213 ^b^ ± 11.5	0.0009	0.0003	0.0579

Groups fed with the standard mixture (T0; control group), or the standard mixture supplemented with 1% (T1), 2% (T2), or 3% (T3) tannin-rich extract Farmatan^®^. * Number of serous acini was counted on eight microscopic fields at 20× magnification. ^a,b^ Different superscript letters within a row indicate significant differences (*p* < 0.05). Values represent mean ± SEM.

**Table 2 animals-12-02171-t002:** Morphological traits of the mandibular glands from fattening boars fed diets supplemented with different tannin levels.

Experimental Group	T0	T1	T2	T3	*p*-Value	Linear Trend	Nonlinear Trend
Area of lobuli [mm^2^]	3.0 ^a^ ± 0.2	4.0 ^ab^ ± 0.2	4.7 ^b^ ± 0.3	3.5 ^a^ ± 0.1	0.0015	0.1816	0.0008
Area of mucous acini [µm^2^]	1314 ± 73	1375 ± 103	1223 ± 101	996 ± 67	0.0794	0.0296	0.3079
Number of mucous acini *	15.3 ± 0.9	13.3 ± 1.7	13.9 ± 2.2	11.3 ± 2.0	0.5162	0.1985	0.7659
Area of mixed acini [µm^2^]	1384 ± 67	1506 ± 67	1414 ± 115	1320 ± 74	0.5276	0.5459	0.4055
Number of mixed acini *	23.9 ± 2.5	28.5 ± 0.5	27.3 ± 0.9	27.4 ± 1.4	0.2245	0.1925	0.2484
Area of ducts [µm^2^]	4055 ± 284	3584 ± 310	3735 ± 233	3011 ± 106	0.1154	0.0381	0.4177
Number of ducts per lobule	6.2 ± 0.7	6.0 ± 0.3	6.7 ± 0.3	7.4 ± 0.5	0.2980	0.1012	0.6299
Area of serous cells [µm^2^]	114.6 ± 7.2	104.9 ± 8.3	109.3 ± 1.7	123.1 ± 11.2	0.2810	0.2222	0.3013
Area of mucous cells [µm^2^]	144.0 ± 3.8	128.1 ± 1.4	155.0 ± 14.8	163.5 ± 14.3	0.3854	0.1826	0.5374

Groups were fed with the standard mixture (T0; control group) or standard mixture supplemented with 1% (T1), 2% (T2), or 3% (T3) tannin-rich extract Farmatan^®^. * Numbers of mucous and mixed acini were counted on eight microscopic fields at 20× magnification. ^a,b^ Different superscript letters within a row indicate significant differences (*p* < 0.05). Values represent mean ± SEM.

## Data Availability

The data presented in this study are available on request from the corresponding author.

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
