# Peer review of "Salivary Gland Adaptation to Dietary Inclusion of Hydrolysable Tannins in Boars"

_animals, 2022, doi:10.3390/ani12172171_

Round 1
Reviewer 1 Report
Dear authors,
your article is interesting. However, some parts should be improved before considering it for publication.
1- Add in the introduction a part on the impact of tannin on growth and production performance, as well as on digestive flora.
2-You use long sentences in your manuscript. For example, lines 89-93 and 137-142. Short, clear sentences should be used.
3-In the Materials and Methods section, you should describe the boxes used in the experiment and the method used to allocate them to the animals in each group.
4- Add the composition of the feed used in the study.
Are the feeds iso-energetic and iso-proteic? If not, are the results not biased?
5-For the statistical analysis, have you checked the normality of the data? I strongly doubt that variables such as: are normally distributed. It would be appropriate to check the normality of the variables and use the appropriate statistical tests.
6-The discussion should be updated with the above remarks.
Best regards.
Author Response
Response: We thank the reviewer for such a positive opinion about our manuscript.
Point 1: Add in the introduction a part on the impact of tannin on growth and production performance, as well as on digestive flora.
Response 1: We thank the reviewer for this suggestion. The introduction paragraph has been updated accordingly to provide sufficient background and include all relevant references.
Point 2: You use long sentences in your manuscript. For example, lines 89-93 and 137-142. Short, clear sentences should be used.
Response 2: The sentence in lines 89 was shortened accordingly. The sentence in lines 137 – 142 includes the list of measured histologic parameters and is required in our opinion, we have also checked for other longer sentences within the manuscript to shorten where possible.
Point 3: In the Materials and Methods section, you should describe the boxes used in the experiment and the method used to allocate them to the animals in each group.
Response 3: The information was included in the Materials and methods section but has now been more adequately described.
Point 4: Add the composition of the feed used in the study.
Response 4: The ingredients (%) and chemical composition of the feed mixtures were described previously (Čandek-Potokar et al., 2015), and the basal feed mixture used (T0) contained 13.2 MJ metabolizable energy and 17.4% protein. The treatment diets were supplemented with 1, 2 and 3 % tannin extract Farmatan in T1, T2 and T3, respectively.
Here is the composition of the feed, which as described previously (Čandek-Potokar et al., 2015).
Table 1. Ingredients and chemical composition of experimental feed mixtures (Čandek-Potokar et al. 2015)
|
|
T0 (Control) |
T1 (1%) |
T2 (2%) |
T3 (3%) |
|
Ingredients [%] |
|
|||
|
Maize |
62.0 |
|||
|
Soya meal |
13.0 |
|||
|
Wheat meal |
8.0 |
|||
|
Rapeseed meal |
7.0 |
|||
|
Sunflower meal |
5.0 |
|||
|
Molasses |
2.0 |
|||
|
Calcuim carbonate |
1.1 |
|||
|
Lysine |
1.0 |
|||
|
Methionine |
0.3 |
|||
|
Monocalcium phosphate |
0.17 |
|||
|
Chemical composition |
|
|
|
|
|
Farmatan [%] |
0.0 |
1.0 |
2.0 |
3.0 |
|
DM§ [g/kg] |
892.5 |
884.8 |
881.4 |
884.9 |
|
Crude protein [g/kg SS] |
174.5 |
168.7 |
166.4 |
164.4 |
|
Ether extract [g/kg SS] |
25.9 |
28.7 |
28.2 |
26.7 |
|
Crude fibre [g/kg SS] |
51.5 |
46.6 |
49.4 |
50.0 |
|
Crude ash [g/kg SS] |
47.6 |
45.4 |
46.5 |
45.2 |
|
ME¤ [MJ/kg] |
13.2 |
13.3 |
13.1 |
13.2 |
- Dry matter, ¤ metabolisable energy
Point 5: Are the feeds iso-energetic and iso-proteic? If not, are the results not biased?
Response 5: We thank the reviewer for this important question. The feed mixtures were app. iso-energetic and iso proteic – the slight deviation/differences of analytical values (see Čandek-Potokar et al., 2015) are due to the supplementing basic feed (T0) with tannin extract in T1, T2 and T3 feed (1 to 3 % respectively) and analytical uncertainty.
Point 6: For the statistical analysis, have you checked the normality of the data? I strongly doubt that variables such as: are normally distributed. It would be appropriate to check the normality of the variables and use the appropriate statistical tests.
Response 6: The statistical evaluation of the results showed normal distribution of the data for all statistically evaluated results (confirmed by Shapiro-Wilk test).
Point 7: The discussion should be updated with the above remarks.
Response 7: The discussion has been updated accordingly to include the aforementioned remarks.
Point 8: Extensive editing of English language and style required.
Response 8: We have used professional English editing service Editage and a native English-speaking colleague and author to improve the English language within the manuscript.
Reviewer 2 Report
I consider the present manuscript acceptable to be published in the renowned Animals Journal because of its relevance to understand a mechanism not properly investigated in pigs. Congratulations to the authors.
The main question is addressed to if there are any modifications in salivar glands of boards fed with a diet containing different levels of hydrolizable tannins (HT). The relevance of the study is undeniable. I consider the topic original and relevant in the field because at as far as I could concluded from reading the article so far there were not other studies dealing specifically with the main subject of the article.
I think valuable information was brougth to light mainly if it is considered that now the positive effects (antimutagenic, antimutagenic, anticancerogenic, antidiarrheal, and antiulcerogenic) of HT can be directed to act in benefit of fattening boars, since now it became clear that they can adapt themselves to digest those substances by modifying their salivar glands.
Specific improvements for authors consideration regarding methodology: A specific methodology to assess in how extent the salivar glands of boars are affected by a diet containing HT several levels of HT.
Conclusions are consistent with the evidence and arguments presented, and they address the main question posed. No doubt about it.
The references are appropriate.
Author Response
Response: We thank the reviewer for such an encouraging opinion about our manuscript. We have extended the methodology in a few places to further enhance methodology explanations.
Reviewer 3 Report
To persuade the readers, please include previous works that demonstrated the beneficial effects of HT addition in pig diets.
Materials and methods: Why did the authors not sample saliva to measure PRP and saliva composition?
Figures 4 and 6 show the results. Please double-check the error bar and superscripts that did not match, as well as the significance level.
Please consider and add these points to the discussion section due to a lack of evidence to confirm a beneficial effect of HT addition on boar production.
Author Response
Point 1: To persuade the readers, please include previous works that demonstrated the beneficial effects of HT addition in pig diets.
Response 1: As thoroughly reviewed by Caprarulo et al (2021), the beneficial effects of tannin supplementation in pig farming are related to their antimicrobial, antioxidant and radical scavenging, anti-inflammatory activities and on the immune status however, the underlying mechanisms are not fully understood.
Point 2: Materials and methods: Why did the authors not sample saliva to measure PRP and saliva composition?
Response 2: The original study included the collection of multiple organs to examine the effect of tannins on pigs. At the time of organ collection, we measured only the weight of salivary glands. After seeing the effect on them, we decided to gain deeper insight and therefore performed the histological analysis.
Point 3: Figures 4 and 6 show the results. Please double-check the error bar and superscripts that did not match, as well as the significance level.
Response 3: Statistical evaluation on Figures 4 and 6 has been reviewed and updated accordingly.
Point 4: Please consider and add these points to the discussion section due to a lack of evidence to confirm a beneficial effect of HT addition on boar production.
Response 4: We thank the reviewer for this comment. This paper presents the results on histology of salivary glands and not specifically on production performance, however we have added extra points to the discussion.
Reviewer 4 Report
Manuscript entitled „Salivary gland adaptation to dietary inclusion of hydrolysable tannins in boars” is an interesting, well-written and well-planned experimental work. However, the text needs some corrections according to the following comments:
Abstract
Line 29 – put the before significant
Introduction
Line 54 – use abbreviation HT instead of full name hydrolysable
Line 56 – should be “are the most” instead is
Line 89 - replace citation Bee et al. 2017) with an appropriate number of citation
Materials and Methods
Line 121 – explain the full name of abbreviation SF
Line 134-135 - explain what these antibodies are anti-PRB2 antibody and anti-PRH1/2 antibody
Line 138 - order one magnification marking scheme x magnification differs from position to position
Line 139 – remove of after all
Discussion
Line 287 – should be: has been addressed
Line 295 – should be: mentioned above
Line 299 – remove who and put receiving after pigs
Line 305 – should be: classify
Line 347 – use abbreviation HT instead of full name
References
Line 388 - complete the parameters for item 3 of the literature
organize the record of journal names - they should be abbreviated
Author Response
Response: We thank the reviewer for such a positive opinion about our manuscript.
Abstract
Point 1: Line 29 – put the before significant
Response 1: We have now put the before significant.
Introduction
Point 2: Line 54 – use abbreviation HT instead of full name hydrolysable
Response 2: We have used abbreviation HT instead of full name hydrolysable.
Point 3: Line 56 – should be “are the most” instead is
Response 3: We have used are the most.
Point 4: Line 89 - replace citation Bee et al. 2017) with an appropriate number of citation
Response 4: We have replaced citation Bee et al. 2017 with a number now.
Materials and Methods
Point 5: Line 121 – explain the full name of abbreviation SF
Response 5: After thorough review of available literature and consultation with Merck technical service, we were unable to find out the full name of abbreviation SF. Light Green SF is a name for a dye included in Merck’s Masson-Goldner staining kit and was used to stain collagen.
Point 6: Line 134-135 - explain what these antibodies are anti-PRB2 antibody and anti-PRH1/2 antibody
Response 6: We have added an explanation for antibodies anti-PRB2 and anti-PRH1/2
Point 7: Line 138 - order one magnification marking scheme x magnification differs from position to position
Point 8: Line 139 – remove of after all
Response 8: We have removed of after all.
Discussion
Point 9: Line 287 – should be: has been addressed
Response 9: We have written has been addressed.
Point 10: Line 295 – should be: mentioned above
Response 10: We have written mentioned above.
Point 11: Line 299 – remove who and put receiving after pigs
Response 11: We have removed who and put receiving after pigs.
Point 12: Line 305 – should be: classify
Response 12: We have written classify.
Point 13: Line 347 – use abbreviation HT instead of full name
Response 13: We have used abbreviation HT instead of full name.
References
Point 14: Line 388 - complete the parameters for item 3 of the literature
Response 14: We have completed the parameters for item 3 of the literature.
Point 15: Organize the record of journal names - they should be abbreviated
Response 15: We have abbreviated journal names.
Round 2
Reviewer 1 Report
Dear authors,
thank you for the improvements made to your article.
I wish you all the best for the future.
Best regards,